# A Tumor Homing Peptide-Linked Arsenic Compound Inhibits Pancreatic Cancer Growth and Enhances the Inhibitory Effect of Gemcitabine

**DOI:** 10.3390/ijms252111366

**Published:** 2024-10-22

**Authors:** Hong He, Chelsea Dumesny, Judith A. Carrall, Carolyn T. Dillon, Katja I. de Roo, Mal Eutick, Li Dong, Graham S. Baldwin, Mehrdad Nikfarjam

**Affiliations:** 1Department of Surgery, University of Melbourne, Studley Road, Heidelberg, VIC 3084, Australia; watsoncj@unimelb.edu.au (C.D.); li.dong@unimelb.edu.au (L.D.); gsbaldwin49@gmail.com (G.S.B.); m.nikfarjam@unimelb.edu.au (M.N.); 2School of Chemistry and Molecular Bioscience, University of Wollongong, Sydney, NSW 2522, Australia; jcarrall@uow.edu.au (J.A.C.); carolynd@uow.edu.au (C.T.D.); kdr964@uowmail.edu.au (K.I.d.R.); 3Phebra Pty. Ltd., Lane Cove West, Sydney, NSW 2066, Australia; mal.e@phebra.com

**Keywords:** arsenic trioxide, peptide-linked arsenic compound, PhAs-LHP, pancreatic cancer, gemcitabine, tumor homing peptide

## Abstract

Arsenic trioxide (ATO) has been shown to inhibit pancreatic cancer (PC) cell growth *in vitro* and to promote the inhibitory effects of gemcitabine (Gem) on PC *in vivo*. However, the high toxicity of ATO associated with the required high doses and indiscriminate targeting has limited its clinical application. This study aimed to determine whether coupling arsenic to a tumor homing peptide would increase the inhibitory potency against PC cells. The effects of this peptide-linked arsenic compound (PhAs-LHP), the analogous non-targeting arsenic compound (phenylarsine oxide, PAO), and marketed ATO on PC growth were tested *in vitro* and in a mouse model. The data demonstrated that PhAs-LHP inhibited PC cell growth *in vitro* more potently, with IC_50_ values 10 times lower than ATO. Like ATO, PhAs-LHP induced cell death and cell cycle arrest. This cytotoxic effect of PhAs-LHP was mediated via a macropinocytosis-linked uptake pathway as amiloride (a macropinocytosis inhibitor) reduced the inhibitory effect of PhAs-LHP. More importantly, PhAs-LHP inhibited PC growth in mice and enhanced the inhibitory effect of Gem on PC growth at 2 times lower molar concentration than PAO. These results indicate that PhAs-LHP inhibited PC more potently than ATO/PAO and suggest a potential clinical use for the combination of Gem with the peptide-linked arsenic compound for the treatment of pancreatic cancer.

## 1. Introduction

The incidence of pancreatic cancer (PC) is rising, with >90% mortality largely owing to the lack of an effective treatment [1]. Surgical resection may be curative but is only applicable to <20% of patients at diagnosis [2]. This leaves over 80% of patients with only a palliative treatment (mainly chemotherapy) option, with a median survival <1 year, regardless of the use of various therapy regimens [3,4]. Gemcitabine (Gem)-based therapies, either alone or in combination with agents including nab-paclitaxel [5], are favored approaches for the treatment of patients with locally advanced or metastatic PC. Although the combination of folinic acid, 5-fluorouracil, irinotecan, and oxaliplatin (FOLFIRINOX) increases survival compared to treatment with gemcitabine alone, it often cannot be tolerated due to the high toxicity and poor performance status of patients [6]. There is an urgent need to develop more effective approaches that can improve the treatment of PC to increase patients’ survival.

Inorganic arsenic, specifically arsenic trioxide (ATO), has been recognized as a toxic compound [7] and a medically effective agent. Although this arsenic compound has been used as a drug to treat various diseases in traditional medicine, its FDA approval in the year 2000 and subsequent successful application for the treatment of acute promyelocytic leukemia (APL) [8,9] triggered research into its role in solid tumor therapy. ATO has been shown to inhibit the growth of solid tumors including breast, hepatic, esophageal, gastric, pancreatic, ovarian, and prostate carcinomas [10], generally by induction of apoptosis. However, the lower specificity of ATO towards the cellular targets in these cancers and the requirement to administer higher doses have become the major causes leading to the failure of the clinical trials using ATO as a single reagent in patients with breast, hepatic, and pancreatic cancers [11,12].

To develop novel strategies for the treatment of PC, the effects of ATO, alone and in combination with other reagents, on PC have been explored. ATO inhibited PC cell growth by inducing apoptosis [13] via downregulation of the S-phase kinase-associated protein 2 (Skp2) [14]. ATO also sensitized the response of PC cells to Gem by inhibition of Skp2. ATO promoted the inhibitory effects of either Gem [15] or the hypoxia-inducible factor 1 inhibitor PX-478 [16] on PC. However, the high toxicity of ATO limited its clinical application as an anticancer agent and contributed to the failure of a clinical trial using ATO for the treatment of patients with pancreatic adenocarcinoma refractory to Gem [12]. The difference between the promising results in pre-clinical mouse models and the failure in clinical trials on patients with PC may have been caused by several factors including the different doses used in mice (5–8 mg/kg) [15,16,17] and in humans (300–350 μg/kg) [12].

The leukemia-homing peptide CAYHRLRRC (LHP) has been shown to target the macropinocytotic pathway in leukemia and lymphoma cells [18]. The presence of cysteine residues at both the *N*- and *C*-termini of the peptide have enabled the synthesis of a cyclic arsenic derivative that could enter cells via macropinocytotic-type pathways [7]. Recognizing that 95% of PC cases exhibit mutant KRAS protein [19] and that KRAS-transformed cells have been shown to have enhanced macropinocytosis [20], it was anticipated that the peptide-linked arsenic compound might also target pancreatic cancers and increase the inhibition of ATO. To test the hypothesis that the peptide-linked arsenic compound could reduce indiscriminate toxicity and increase anticancer efficacy compared to ATO, the peptide-linked phenylarsine compound (herein referred to as PhAs-LHP) was prepared and characterized as described in our previous publication [7], and its effects on PC growth were assessed *in vitro* and *in vivo*.

## 2. Results

### 2.1. ATO Suppressed Pancreatic Cancer Cell Growth, and This Inhibitory Effect Was Enhanced by Gemcitabine and PF-3758309

Human (PANC-1 and MiaPaCa-2) and murine (KPCWT942 and TB33117) pancreatic cancer (PC) cells were treated with ATO (1–50 μM) for 24, 48, and 72 h. Cell proliferation was measured by the MTT assay, and the readouts from the untreated control cells were taken as 100%. ATO decreased the proliferation of all four PC cell lines in a dose-dependent manner (Figure 1). The maximal inhibitory effects of ATO were reached after 48 h of incubation (Table 1), when the minimal IC_50_ values were reached in most cell lines.

Gem is used as a standard treatment for PC, and tremendous efforts have been made to develop Gem combination therapies aimed at improving therapeutic outcomes and increasing patients’ survival. PF-3758309 is a chemical inhibitor that targets p-21-activated kinases (PAKs) and is specific to PAK4 [21]. PAKs, particularly PAK1 and PAK4, play critical roles in PC progression [22]. Inhibition of PAK4 by PF-3758309 has enhanced the sensitivity to multiple chemotherapeutic reagents in patient-derived PC cell lines [23]. To determine the effects of ATO in combination with Gem or PF-3758309, both human and murine PC cells were pre-incubated with Gem or PF-3758309 for 24 h, followed by 48 h incubation with ATO at the concentrations indicated in Figure 2. The observation that pre-treatment of PC cells with either Gem or PF-3758309 enhanced the inhibitory effects of ATO on PC cell growth to different degrees (Figure 2) indicated that the combination of ATO with either Gem or PF-3758309 suppressed PC growth more potently than treatment with either single reagent.

### 2.2. Peptide-Linked Arsenic Compound Inhibited Pancreatic Cancer Cell Growth More Potently than ATO via Regulating Cell Cycle and Cell Death

To facilitate its entry into cells, arsenic was coupled to the leukemia homing peptide CAYHRLRRC (LHP) by reacting the peptide with phenylarsine oxide (PAO) to generate PhAs-LHP [7]. The effect of PhAs-LHP on PC cell growth was then compared with LHP and ATO. Human (PANC-1 and MiaPaCa-2) and murine (TB33117) PC cells were treated with ATO, PhAs-LHP, or free peptide LHP for 48 h, and cell proliferation was measured by the MTT assay. PhAs-LHP inhibited the proliferation of human (MiaPaCa-2 and PANC-1) and murine (TB33117) PC cells (Figure 3), and LHP did not affect the proliferation of these PC cells (Figure 3). More importantly, PhAs-LHP suppressed the proliferation of both human and murine PC cells more potently compared to ATO, as the IC_50_ values obtained for PhAs-LHP were ten times less than the IC_50_ values for ATO, as shown in Table 2. These results suggested that coupling to the small peptide LHP facilitated the entry of arsenic into cells and hence promoted its inhibition of cell proliferation. While the highest IC_50_ of ATO on PC cells was less than 4 μM, both ATO and PhAs-LHP had only reduced the proliferation of a normal pancreatic epithelial cell line (HPDE) by 10% at 5 μM, and there was no difference in inhibition between ATO and PhAs-LHP up to 5 μM, as shown in Appendix A.

ATO inhibited PC cell growth by stimulation of apoptosis [16,24]. The effects of PhAs-LHP on cell cycle and cell death were determined by flow cytometry analysis. In MiaPaCa-2 cells, after 48 h treatment, ATO induced about 15% cell death at 0.5 μM while PhAs-LHP (0.05 μM) caused 24% cell death at ten times lower concentrations than ATO (Figure 4A). Likewise, after 24 h treatment, ATO (2 μM) and PhAs-LHP (0.2 μM) induced cell cycle arrest at the G2/M phase in PANC-1 cells (Figure 4B). These results demonstrated the intracellular mechanism for the more potent inhibition of this small peptide-linked compound on PC growth compared to ATO (Figure 3).

### 2.3. PhAs-LHP Exerted Its Cytotoxic Effect via Macropinocytotic Uptake of LHP Ligand in PC Cells

Before *in vivo* studies of PhAs-LHP in the PC model, it was important to establish that the PC cell line used for the xenograft was (1) targeted by the ligand, LHP, using the macropinocytotic uptake pathway and (2) that the toxicity of PhAs-LHP towards the cell line employed a macropinocytotic uptake mechanism. Evidence of each of these instances would be supportive of/consistent with the feasibility of the targeting mechanism of the LHP ligand in vivo.

The uptake of the fluorescent 5FAM-LHP was evaluated in live MiaPaCa-2 cells. Figure 5a shows a typical confocal image of the uptake of 5FAM-LHP into MiaPaCa-2 cells counterstained with Hoechst 33345 following 1 h incubation. The level of uptake was quantified as the total particle area/total cell area (Figure 5b). Following 1 h incubation, bright green punctate was observed in the cytoplasm of MiaPaCa-2 cells (Figure 5a), with the corresponding total particle area/total cell area calculated as 10.3 ± 1.6 (Figure 5b). When co-treated with 25 µM amiloride (a macropinocytosis inhibitor) for 1 h, the uptake of 5FAM-LHP into MiaPaCa-2 cells decreased by 32% (7.05 ± 0.60, Figure 5b, *p* < 0.01). A greater decrease (~71%) in 5FAM-LHP uptake was observed following a 16 h pre-treatment of MiaPaCa-2 cells with 25 µM amiloride before 1 h incubation with 5FAM-LHP (3.0 ± 0.7, Figure 5b, *p* < 0.0001) compared to cells incubated with 5FAM-LHP alone. Furthermore, the effects of ATO/PAO and PhAs-LHP on MiaPaCa-2 proliferation with or without the macropinocytosis inhibitor, amiloride (25 μM), were determined. By 24 h incubation, amiloride did not affect the IC50 values of ATO or PAO (non-targeting arsenic compounds), as shown in Figure 5c. In contrast, inhibition of macropinocytosis by amiloride significantly increased the IC50 value of PhAs-LHP about two times its IC50 in the absence of amiloride (Figure 5c, Table 3), indicating that the cytotoxicity of PhAs-LHP was reduced when the macropinocytosis uptake pathway was inhibited. This suggests that PhAs-LHP used the macropinocytosis pathway to enter MiaPaCa-2 cells.

### 2.4. Arsenic Peptide Suppressed Pancreatic Cancer Growth and Enhanced the Inhibitory Effect of Gemcitabine on Pancreatic Cancer Growth

To determine the effects of PhAs-LHP on PC growth *in vivo*, human PC cells (MiaPaCa-2) were subcutaneously injected into the flanks of Scid mice, which were then treated with phenylarsine oxide (PAO, 50 μg/kg) or PhAs-LHP (200 μg/kg) by intraperitoneal injection as described in the Materials and Methods section. The doses of PAO and PhAs-LHP were calculated and adjusted with reference to the results obtained from the in vitro assay. PAO was selected as a control because it is the As-containing product produced following the dissociation of LHP from PhAs-LHP. Both PAO and PhAs-LHP significantly suppressed pancreatic tumor growth in Scid mice, as demonstrated by a reduction in tumor volumes (Figure 6A) during the treatment period and decreased tumor weight at the end of the treatment (Figure 6B). The mouse body weights were slightly and consistently increased throughout the treatment period (Figure 6D). There was no significant difference in either tumor volumes or tumor weight between PAO- and PhAs-LHP-treated mice. Although the dose of PhAs-LHP was four times greater than the dose of PAO based on weight, the eight-fold greater molecular weight of PhAs-LHP (1368.2 g/mol) compared to PAO (168.0 g/mol) suggests that PhAs-LHP was more potent than PAO, as this effectively equates to half the amount of arsenic in the PhAs-LHP dose administered to the mice. The results showed that PAO and PhAs-LHP suppressed PC growth in Scid mice to a similar degree at the dosages used in this experiment, where the molar concentration of PhAs-LHP was two times less than PAO. These results suggested that the peptide-linked arsenic compound suppressed the pancreatic cancer growth *in vivo* more potently than PAO.

The *in vitro* data showed that the combination of ATO and Gem inhibited PC cell proliferation more potently than either compound singly (Figure 2). To determine if the combination of PhAs-LHP and Gem suppressed PC growth *in vivo* more effectively than either compound singly, human PC cells (MiaPaCa-2) were subcutaneously injected into the flanks of Scid mice, which were then treated with Gem alone (50 mg/kg) or with gemcitabine (50 mg/kg) plus PAO (50 μg/kg) or PhAs-LHP (200 μg/kg) by intraperitoneal injection as described in the Materials and Methods section. Gem alone significantly decreased PC growth, as shown by reduced tumor volumes (Figure 7A) and tumor weight (Figure 7B). The apparent further reduction in tumor volumes and weight by PAO on top of the Gem did not reach statistical significance. However, the peptide-linked arsenic compound, PhAs-LHP, further decreased tumor volumes (Figure 7A) and tumor weight (Figure 7B) significantly when combined with Gem treatment. The results showed that PhAs-LHP enhanced the inhibitory effect of Gem on PC growth *in vivo*. The tumor volume and weight of Gem plus PhAs-LHP were lower than those of Gem plus PAO, though they did not reach statistical significance. However, the molar concentration of PhAs-LHP was two times less than PAO, indicating the more potent inhibition by PhAs-LHP. When comparing the relative tumor growth by taking the tumor volumes measured on Day 7 (when the treatment started) as 100%, the tumor growth of Gem plus PhAs-LHP was significantly slower than of Gem plus PAO on Day 27 (Appendix A). In the presence of Gem, the inhibitory effects of both PAO and PhAs-LHP were increased compared to PAO or PhAs-LHP alone, respectively (Appendix A).

No toxicity was observed with PAO, PhAs-LHP, or Gem with the dosages used, as demonstrated by the absence of any significant decrease in mouse body weight (Figure 7D). Furthermore, the arsenic distribution was determined to the brain, heart, liver, and kidney of the mice from the non-treated control, Gem, Gem + PAO, and Gem + PhAs-LHP groups, as shown in Figure 8. The tissues of these organs were analyzed by graphite furnace atomic absorption spectrophotometry (GFAAS) to detect if there were any differences in the arsenic (As) distribution between the two arsenic treatment groups (PAO or PhAs-LHP). No As was detected (as defined by the GFAAS detection limit) in any organ samples from the control or the Gem treatment group. The only significant As detected was in the hearts and the livers of the PAO + Gem-treated mice (Figure 8a, *p* < 0.0001), although there was no detectable As in the brains or the kidneys of this treatment group. The As concentration in the livers was twice that in the hearts (Figure 8, *p* < 0.0001). For the PhAs-LHP + Gem-treated mice, As was not detected in any of the tissue samples. This is consistent with the specific targeting of PhAs-LHP towards the pancreatic tumor.

### 2.5. Arsenic Peptide Suppressed Pancreatic Cancer Growth by Downregulation of Skp2

Previous reports have demonstrated that ATO inhibited pancreatic xenografted tumor growth by decreasing proliferation and increasing apoptosis *in vivo* [17,25] and that ATO suppressed PC cell proliferation and migration by downregulation of S-phase kinase-associated protein 2 (Skp2) [14], a cell cycle regulator playing oncogenic roles in cell growth, apoptosis, migration/invasion and angiogenesis [26], and chemoresistance [14,27,28]. To explore the mechanism involved in the effects of ATO and its peptide-linked compound on pancreatic tumors, Skp2 expression in xenografted tumors was determined by immunohistochemical staining. The overall levels of Skp2 detected were low. The levels of Skp2 were reduced in the tumors of the mice treated with PAO, PhAs-LHP, and gemcitabine, respectively (Figure 9), compared to the non-treatment control. The combination of gemcitabine and PAO or PhAs-LHP did not further reduce the expression of Skp2 compared to single reagent treatments (Figure 9).

## 3. Discussion

The observation that, despite its effective inhibition of PC cells in vitro, ATO has not shown any activity in PC patients with progressive disease after Gem treatment [12] indicates that new approaches are required. To reduce the toxicity and increase the anticancer efficacy of ATO, a peptide-linked arsenic compound (PhAs-LHP, where LHP is a peptide that possesses a penetrating motif (RLRR) that promotes uptake by a macropinocytotic pathway that is upregulated in KRAS-transformed cells) was synthesized, and its effect was tested on pancreatic cancer growth *in vitro* and *in vivo* and compared to the effects of the analogous non-peptide-linked arsenic compounds PAO and ATO. The peptide-linked arsenic compound inhibited PC cell growth more potently, with IC_50_ values ten times lower than ATO. More importantly, the peptide-linked arsenic compound inhibited PC growth in mice and enhanced the inhibitory effect of Gem on pancreatic tumour growth at a two times lower molar concentration of PAO.

The peptide-linked compound, PhAs-LHP, induced cell death in MiaPaCa-2 cells at ten times lower concentrations than ATO (Figure 4A). Similarly, PhAs-LHP caused G2/M cell cycle arrest of PANC-1 cells at ten times lower concentrations than ATO (Figure 4B). These findings provide an intracellular mechanism for the more potent inhibitory effect of PhAs-LHP, compared to ATO, on PC cell growth and are also consistent with previous reports that ATO induces apoptotic cell death in PC cells via changes in the cell cycle [13,14].

The cytotoxic effect of PhAs-LHP was inhibited by the macropinocytotic inhibitor, amiloride, in MiaPaCa-2 cells while that of ATO and PAO was unaffected. This reinforces the proposal that PhAs-LHP targets the PC cells via a macropinocytotic pathway.

Consistent with the *in vitro* findings, the peptide-linked compound PhAs-LHP suppressed xenografted PC growth in Scid mice more potently than the non-peptide-linked arsenic (PAO), with two times lower molar concentration (Figure 6), and further enhanced the inhibitory effect of Gem on PC growth in vivo (Figure 7). PAO alone did not significantly increase the inhibition of Gem on PC growth in vivo (Figure 7), although ATO was shown to promote the inhibitory effect of Gem in a similar xenografted PC growth model where a higher dosage of ATO was used [14]. The difference between the data presented here and the previous report [14] may be due to the different concentrations of ATO (5 mg/kg [14]) or PAO (50 μg/kg) used in the experiments. The fact that PhAs-LHP, at two times lower molar concentrations than PAO, enhanced the inhibitory effect of Gem on PC growth *in vivo* while PAO did not increase the inhibitory effect of Gem significantly indicates that PhAs-LHP sensitizes PC response to Gem more potently. Thus, the peptide-linked arsenic compound acts more effectively alone and in combination with Gem to suppress PC growth.

The reduction in tumor volume was similar between PAO + Gem and PhAs-LHP + Gem, although the tumor weight of the mice treated with PhAs-LHP showed minimal but significant reduction compared to PAO + Gem (Figure 7). The arsenic distribution in the heart was much lower in the PhAs-LHP + Gem-treated mice than in the PAO + Gem-treated mice (Figure 8), leading to a potential reduction in cardiotoxicity, which is an adverse effect observed with arsenic chemotherapy.

The toxicity of ATO has limited its clinical application, contributing to the failure in clinical trials of ATO as a cancer treatment [9]. A low dosage used in a clinical trial in PC patients did not benefit patients at all [12]. The peptide-linked arsenic compound PhAs-LHP demonstrated anti-cancer effects on PC in vitro (Figure 3 and Figure 4, Table 2) and in vivo (Figure 6 and Figure 7) at a lower concentration than ATO, indicating more potent anti-cancer effects of this peptide-linked arsenic compound with reduced toxicity. The results here show that due to the targeting properties imparted by linking the peptide to arsenic, a significant reduction in tumor growth can be obtained with substantially less administration of arsenic, which, in turn, should reduce indiscriminate toxicity. PhAs-LHP, at two times less molar concentration than PAO, reduced tumor growth to a similar level by PAO. The two-times difference may be minimal. However, the fact that there is a lot less arsenic distribution in the heart and liver from the PhAs-LHP-treated mice than from the PAO-treated mice indicated a reduced toxicity from PhAs-LHP. Together, these observations imply a potential clinical application of this peptide-linked arsenic compound on its own or in combination with Gem.

PhAs-LPH, at two-times lower concentration than PAO, showed similar efficacy of inhibiting pancreatic tumor growth as PAO did. The two-times difference may be minimal. The reduced toxicity from PhAs-LHP demonstrated by significantly less arsenic distribution in the heart and liver indicates the potential for further improving the anti-tumor efficacy by increasing the dosage of PhAs-LHP. Modification of the peptide-linked arsenic compound with increased efficacy and reduced toxicity will promote the clinical application of the arsenic-based chemotherapy.

Our findings obtained from the *in vitro* and *in vivo* study are consistent with our previous report that PhAs-LHP increased the cellular uptake of arsenic compound most likely via a macropinocytosis pathway, which is upregulated in many cancer cells, including PC cells [7]. It is known that ATO inhibited pancreatic xenografted tumor growth by decreasing proliferation and increasing apoptosis *in vivo* [17,25]. Furthermore, ATO suppressed PC cell proliferation and migration by downregulation of Skp2 [14]. As well as confirming the previously reported effects of ATO on tumor proliferation and apoptosis, our findings that ATO and its peptide-linked compound PhAs-LHP reduced pancreatic tumor growth by decreasing the expression of Skp2 in tumors have added to the mechanisms involved in the effects of ATO and PhAs-LHP. Further study will be needed to determine how the Skp2 pathway contributes to inhibiting PC growth by ATO and PhAs-LHP. Our findings here also warrant future investigation of the systemic effect of ATO and its peptide-linked compound and of the extent to which this peptide-linked arsenic compound would reduce the toxicity of ATO in vivo.

## 4. Materials and Methods

### 4.1. Cell Culture and Reagents

The human PC cell lines PANC-1 and MiaPaCa-2 were purchased from the American Type Culture Collection. The murine pancreatic cancer cell lines KPCWT942 and TB33117 were isolated and characterized from KPC mice as described previously [29]. Phenasen ATO was provided by Phebra Pty. Ltd. (Lane Cove West, NSW, Australia). PhAs-LHP was prepared and characterized as previously published [7]. PF-3758309 was purchased from Active Biochemical Co. (Maplewood, NJ, USA), and Gem was obtained from Sigma-Aldrich (Sydney, Australia). PAO was purified by HPLC before use by us. LHP was obtained from Peptide 2.0 (Chantilly, VA, USA). Cells were cultured in Dulbecco’s Modified Eagle’s Medium (DMEM) supplemented with 5% fetal bovine serum (FBS) obtained from Hyclone Laboratories (Melbourne, Australia) in a 37 °C incubator with a humidified atmosphere of 5% CO_2_.

### 4.2. Cell Proliferation Assay

Human and murine PC cells were incubated in 5% FBS in DMEM with different concentrations of ATO and PhAs-LHP for the periods indicated in the Results section. For the combination of ATO with Gem or PF-3758309, the cells were pre-incubated with gemcitabine or PF-3758309 for 24 h, followed by incubation with different concentrations of ATO for 24 h. Cell proliferation was then measured by the MTT assay.

### 4.3. Measurements of Cell Death and Cell Cycle

Cell death and cell cycle were assayed using propidium iodide (PI) and flow cytometry. Cells were treated with ATO or PhAs-LHP with the concentrations indicated in the text for 48 h for measuring cell death or 24 h for determining cell cycle arrest. All floating and adhering cells were then trypsinized and washed two times with cold phosphate-buffered saline (PBS). For cell death assays, cells were harvested and resuspended in PBS containing 1 μg/mL PI. For cell cycle analysis, cells were harvested and resuspended in PBS containing 0.1% Triton X-100, 50 μg/mL PI, and 10 μg/mL RNaseA (Thermo Scientific, Melbourne, Australia) and then incubated at 37 °C for 15 min. Samples were processed with the FACSymphony A3 flow cytometer (BD Biosciences), and the acquired data were analyzed with the FlowJo software (BD Biosciences, San Jose, CA, USA).

### 4.4. Investigating the Cellular Mechanism of PhAs-LHP

Confocal microscopy was used to determine whether the fluorescently labeled 5-FAM-LHP peptide enters PC cells via macropinocytosis. MiaPaCa-2 cells (1 × 10^5^ cells/mL) were seeded in a glass-bottom dish and incubated in DMEM with HEPES (25 mM), L-glutamine, and FBS (10%) for 24 h. The cells were then incubated with fluorescent-labeled 5-FAM-LHP (5 μM) with or without amiloride (25 μM) in the dark for 1 h. Alternatively, the cells were pre-incubated with amiloride (25 mM) for 24 h before incubation with 5-FAM-LHP (5 μM) for 1 h in the dark. By the end of incubation, the medium was removed and the cells were incubated with Hoechst 33342 (1 mL, 5 µM) for 10 min in the dark before phenol-red free medium (1 mL, RPMI or DMEM) was added for imaging.

All cell samples were imaged using the Leica Application Suite X (3.5.7.23225) on a Leica TCS SP8 Laser Scanning confocal microscope (Leica Microsystems, Wetzlar, Germany) maintained within a microscopy-appropriate incubation chamber (37 °C, 5% CO_2_). Images were acquired in multi-channel mode using a 63× oil immersion lens, a pinhole of 1AU, and the 405 nm diode (Hoechst; gating: 415–478 nm; gain: 720–910 V, depending on cell line), OPSL 488 nm (FITC-dextran/5FAM-peptide; gating: 488–561 nm; gain: 780–950 V, depending on cell line), and PMT Trans (brightfield; gain: 215–280 V, depending on cell line) lasers at 10% laser power. Z-stacks (step size: 1 µm; x: 350 µm; y: 350 µm; z: 20–50 µm) were captured and merged into single images for analysis using Image J.

MiaPaCa-2 cells were seeded in a 96-well plate and incubated with ATO, PAO, and PhAs-LHP with a range of concentrations with or without amiloride (25 μM) for 24 h. The cell proliferation was assessed by MTT, and IC_50_ values for ATO, PAO, and PhAs-LHP were calculated.

### 4.5. Mouse Study

All mouse experiments were approved by the Austin Health Animal Ethics Committee (A2019/05654). Experimental mice were housed in the BioResource Facility at Austin Health and monitored according to health criteria. Human PC cells MiaPaCa-2 (5 × 10^6^ cells/100 μL culture medium) were subcutaneously injected into the flanks of 6-week-old, male Scid mice. When a tumor reached 50 mm^3^ (approximately 7 days), phenylarsine oxide (PAO, 50 μg/kg) or PhAs-LHP (200 μg/kg) was given by intraperitoneal injection every other day for 4 weeks. The control mice were given the same volume of saline. For combined treatment with gemcitabine, gemcitabine was given by intraperitoneal injection every 4 days for 4 weeks. There were 4 to 8 mice per group. Tumor growth was determined by tumor volume measured with callipers every other day and by tumor weight measured at the end of each experiment.

### 4.6. GFAAS Determination of Arsenic Distribution in Mouse Tissue

The brain, heart, liver, and kidney of the mice from the control, Gem, PAO + Gem, and PhAs-LHP + Gem groups (as described in Section 4.5) were collected, and tissues from these organs were prepared and subjected to graphite furnace atomic absorption spectrophotometry (GFAAS) to determine arsenic distribution and concentrations. The detailed materials and method for GFAAS analysis were described in the Appendix A.

### 4.7. Immunohistochemical Staining

Tumor tissues from the mouse study were formalin-fixed, paraffin-embedded, and cut into a 4 μM slide. For immunohistochemical staining (IHC), antigen was retrieved by boiling the samples in citrate buffer (10 mM citric acid, pH6, Sigma-Aldrich, Castle Hill, NSW, Australia) for 30 min, followed by incubation with hydrogen peroxidase blocker for 15 min and then with 5% normal goat serum for 30 min at room temperature for endogenous peroxidase quenching and protein blocking, respectively. Subsequently, the samples were incubated with an anti-Skp2 antibody (1:400, Bioss, Woburn, MA, USA) at 4 °C overnight, followed by staining with the EnVision kit (Dako, Botany, Australia) and hematoxylin (Sigma-Aldrich). The images of all slides were captured and scanned with an Aperio AT2 digital pathology scanner (Leica Biosystems, Melbourne, Australia). The samples were analyzed using QuPath version 0.4.3. The percentage of positively stained cells was calculated.

### 4.8. Statistical Analysis

All values are expressed as mean ± standard error. The in vitro data are from three independent experiments each in triplicate. The in vivo data were collated according to the number of tumor samples. Data were analyzed by one-way ANOVA (SPSS, IBM, New York, NY, USA). In GFAAS studies and amiloride inhibition studies, statistical analyses were performed using GraphPad Prism 5 with one-way ANOVA followed by a Dunnett Multiple Comparison Test (treatment mean vs control mean), and a Tukey–Kramer Multiple Comparison Test (to compare all means). Differences between two means with *p* < 0.05 were considered significant.

## 5. Conclusions

The peptide-linked arsenic compound suppressed pancreatic cancer more potently than the non-peptide-linked arsenic by inhibiting PC cell growth in vitro with IC_50_ values ten times lower than ATO and by reducing PC growth in vivo at a molar concentration two times lower than PAO. Furthermore, at a two-times lower arsenic concentration, a peptide-linked arsenic acid enhanced Gem’s inhibition of PC. These results imply the potential clinical application of this peptide-linked arsenic compound in cancer treatment with reduced dosage and increased effectiveness.

## 6. Patents

There are 14–15 patents/provisional patents on this drug worldwide; inventors are C. Dillon and J. Carrall.

## Figures and Tables

**Figure 1 ijms-25-11366-f001:**
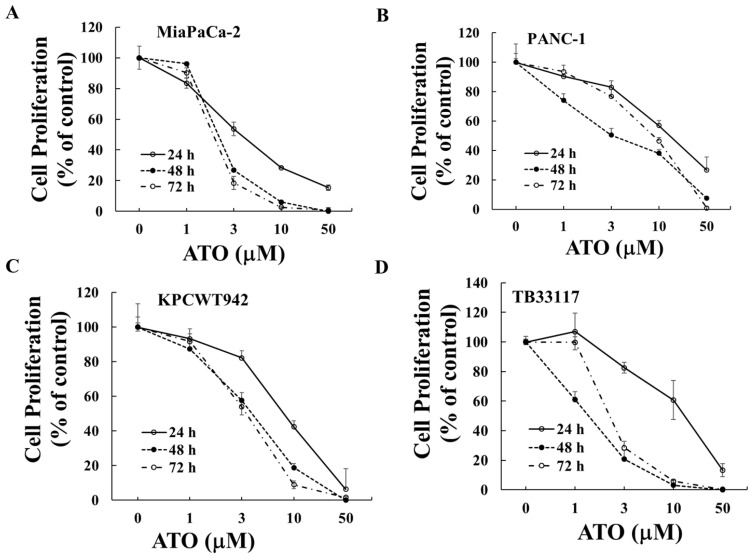
**ATO inhibited pancreatic cancer cell growth.** Human (MiaPaCa-2 (**A**) and PANC-1 (**B**)) and murine (KPCWT942 (**C**) and TB33117 (**D**)) pancreatic cancer (PC) cells were incubated with ATO for 24, 48, and 72 h. Cell proliferation was measured by the MTT assay as described in the Materials and Methods section. The values of control cells without treatment with ATO were defined as 100%. The results were representatives from three independent experiments.

**Figure 2 ijms-25-11366-f002:**
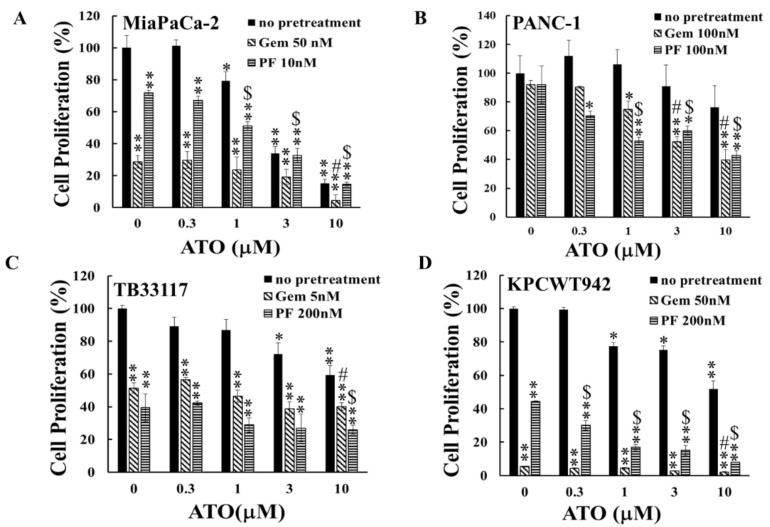
**The inhibitory effect of ATO on pancreatic cancer cell proliferation was enhanced by pre-treatment with gemcitabine or PF-3758309.** Human (MiaPaCa-2 (**A**) and PANC-1 (**B**)) and murine (TB33117 (**C**) and KPCWT942 (**D**)) pancreatic cancer (PC) cells were pre-treated with gemcitabine (Gem) and PF-3758309 (PF) at concentrations indicated in the figures for 24 h, followed by 48 h treatment with arsenic trioxide (ATO). Cell proliferation was measured by the MTT assay. The values of control cells without any treatment were defined as 100%. The results were summarized from three independent experiments. *: *p* < 0.05; **: *p* < 0.01, compared to the non-treated control; #: *p* < 0.05, compared to Gem-pre-treatment only; $: *p* < 0.05, compared to PF pre-treatment only.

**Figure 3 ijms-25-11366-f003:**
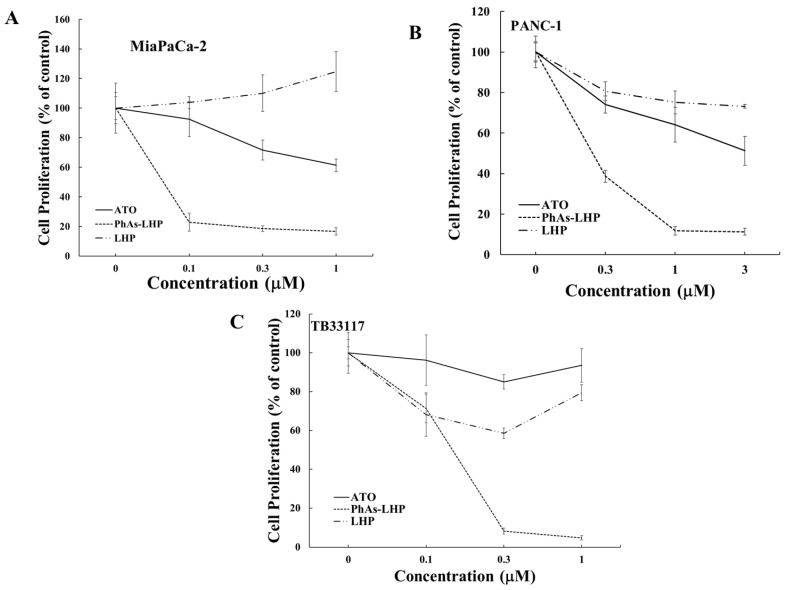
**PhAs-LHP inhibited pancreatic cancer cell growth more potently.** Human (MiaPaCa-2 (**A**) and PANC-1 (**B**)) and murine TB33117 (**C**) pancreatic cancer (PC) cells were incubated with ATO, PhAs-LHP, or the peptide LHP for 48 h. Cell proliferation was measured by the MTT assay. The values of control cells without any treatment were defined as 100%. The results were summarized from at least three independent experiments.

**Figure 4 ijms-25-11366-f004:**
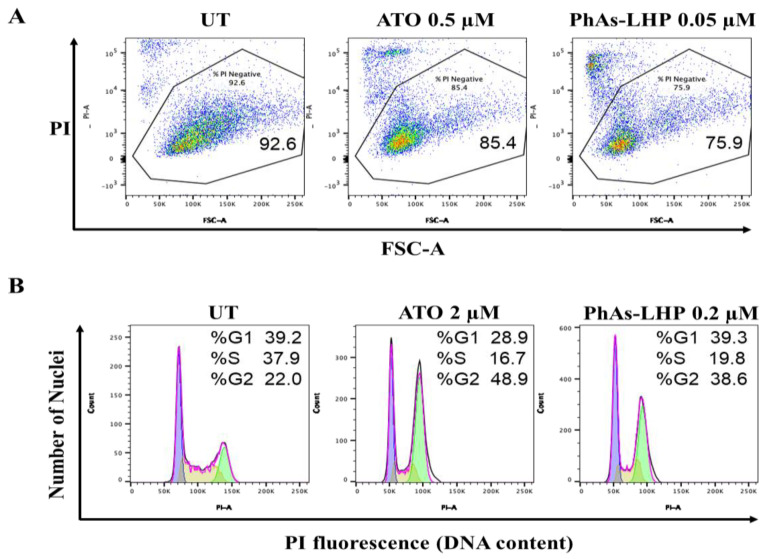
**PhAs-LHP induced cell death and cell cycle arrest more potently than ATO.** MiaPaCa-2 cells were treated with ATO (0.5 μM) or PhAs-LHP (0.05 μM) for 48 h (**A**). PANC-1 cells were treated with ATO (2 μM) or PhAs-LHP (0.2 μM) for 24 h (**B**). Both MiaPaCa-2 and PANC-1 cells were then stained with propidium iodide (PI) and subjected to flow cytometry analysis. Numbers indicate the percent of PI-negative cells of a total of 10,000 cells analyzed per condition (**A**).

**Figure 5 ijms-25-11366-f005:**
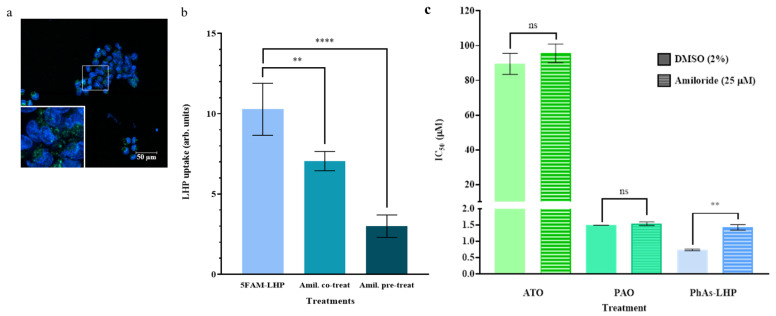
**PhAs-LHP inhibited MiaPaCa-2 proliferation via LHP-mediated macropinocytosis uptake.** Uptake of 5FAM-labeled LHP peptide into live MiaPaCa-2 human cells. (**a**) Typical confocal image of 5FAM-LHP (green) in live cells counterstained with Hoechst 33342 (blue) following incubation (1 h, 37 °C). Image taken using a Leica SP8 microscope at 63× magnification where scale bar = 50 µm. (**b**) Comparison of the uptake of 5FAM-LHP in live MiaPaCa-2 cells following 1 h incubation in the absence or presence of the macropinocytosis inhibitor amiloride (25 µM, 1 h co-treatment or 16 h pre-treatment). Uptake is presented as the mean total particle area/total cell area. Error bars represent the standard deviation from the mean (n = 8–15). (**c**) Comparison of the IC_50_ values obtained from MTT assays of MiaPaCa-2 cells following 24 h treatment with ATO, PAO, or PhAs-LHP alone or in the presence of amiloride (25 µM) for 24 h. Error bars represent the standard deviation from the mean (n = 3). Statistical significance is indicated by **, *p* < 0.01, or ****, *p* < 0.0001, ns, not significant.

**Figure 6 ijms-25-11366-f006:**
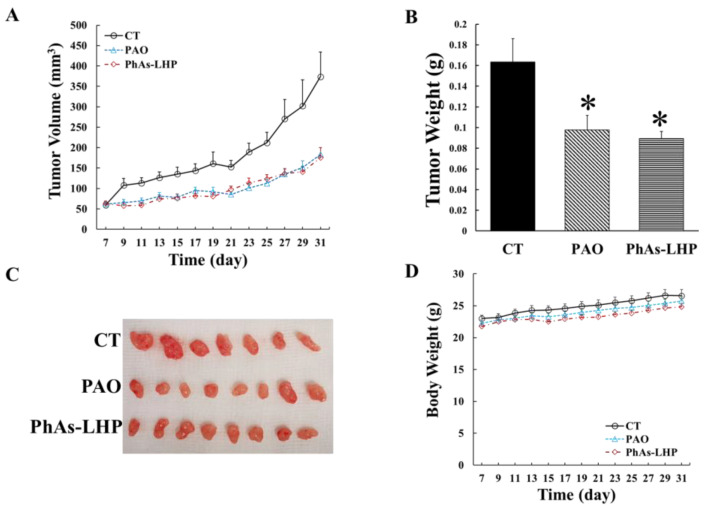
**PhAs-LHP suppressed pancreatic cancer growth in Scid mice.** MiaPaCa-2 human pancreatic cancer cells (5 × 10^6^ cells) were subcutaneously injected into the flank of a Scid mouse. When the tumor size reached >50 mm^3^, PAO (50 μg/kg) or PhAs-LHP (200 μg/kg) was given by intraperitoneal (i.p.) injection every other day for the time periods indicated in (**A**). The control (CT) mice were treated with saline. Tumor volume (**A**) and weight (**B**) were measured as described in the Materials and Methods section. There were 7, 8, and 8 mice in the control, PAO, and PhAs-LHP groups respectively, as indicated in the photo of tumors isolated from each mouse (**C**). The mouse body weight was monitored and shown in (**D**). *, *p* < 0.05, compared to control.

**Figure 7 ijms-25-11366-f007:**
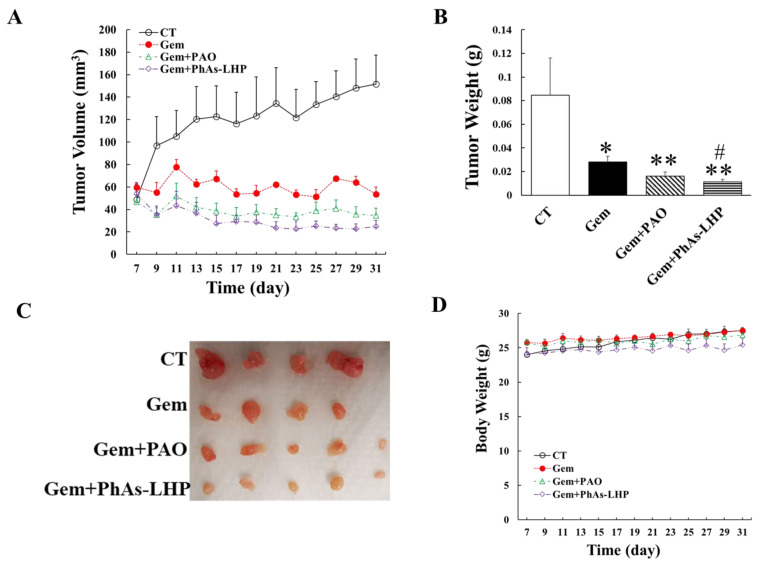
**PhAs-LHP enhanced the inhibitory effect of gemcitabine on pancreatic cancer growth in Scid mice.** Human pancreatic cancer cells, MiaPaCa-2 (5 × 10^6^ cells) were subcutaneously injected into the flank of a Scid mouse. When tumor size reached >50 mm^3^, gemcitabine (Gem, 50 mg/kg), Gem plus PAO (50 μg/kg), or PhAs-LHP (200 μg/kg) was given by intraperitoneal (i.p.) injection for the time periods indicated in (**A**). Gem was given every 4 days. PAO and PhAs-LHP were given every other day. Tumor volume (**A**) and weight (**B**) were measured as described in the Materials and Methods section. There were 4, 4, 5, and 5 mice in the control, Gem, Gem + PAO, and Gem + PhAs-LHP groups, respectively, as indicated in (**C**). The mouse body weight was monitored and shown in (**D**). CT: control; *, *p* < 0.05; **, *p* < 0.01, compared to control; #, *p* < 0.05, compared to Gem alone.

**Figure 8 ijms-25-11366-f008:**
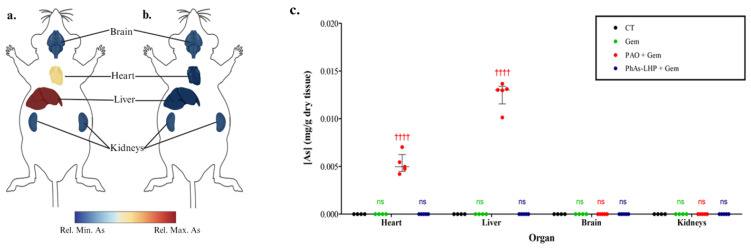
**Arsenic distribution of PAO and PhAs-LHP.** The brain, heart, liver, and kidney tissues from the mice of the control, Gem, PAO + Gem, and PhAs-LHP + Gem groups were analyzed by graphite furnace atomic absorption spectrophotometry (GFAAS) to determine the arsenic (As) concentrations (**c**). Heat maps showed the As distributions in these organs: (**a**) PAO + Gem; (**b**) PhAs-LHP + Gem. Colors represent the relative mean concentrations of As detected. Rel. Min. As, relative minimum amount of As detected, Rel. Max. As, relative maximum amount of As detected. Heat map figures (**c**) created in BioRender.com. (29 September 2024) The treatment groups are represented by: control group (black, n = 4), gemcitabine treatment (green, n = 4), PAO + gemcitabine (red, n = 5), and PhAs(LHP) + gemcitabine (blue, n = 5). Each data point represents the specified data from one mouse. Statistical significance with respect to the control group is indicated by: ns (not significant), †††† (*p* < 0.0001). Data are represented as the mean with the interquartile range.

**Figure 9 ijms-25-11366-f009:**
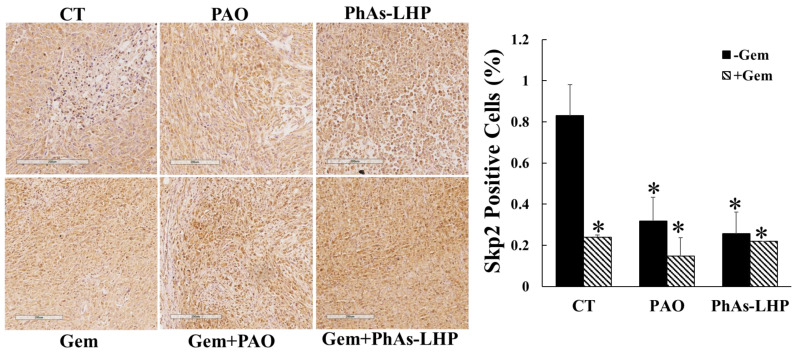
**The expression of Skp2 was reduced by PAO, PhAs-LHP, and gemcitabine.** Tumor tissues from the experiments shown in Figure 5 and Figure 6 were immunohistochemically stained by Skp2. The expression of Skp2 was decreased in the tumors of mice treated with PAO, PhAs-LHP, and gemcitabine. The combination of gemcitabine with PAO or PhAs-LHP did not further decrease the levels of Skp2. *, *p* < 0.05 compared to the untreated control. CT, control; Gem, gemcitabine.

**Table 1 ijms-25-11366-t001:** IC_50_ of ATO (μM) inhibition of PC cell proliferation.

	24 h	48 h	72 h
**MiaPaCa-2**	3.4 ± 0.5	2.0 ± 0.3	1.9 ± 0.3
**PANC-1**	13.8 ± 1.1	3.7 ± 0.6	8.4 ± 0.9
**KPCWT942**	7.9 ± 0.9	3.6 ± 0.6	3.2 ± 0.5
**TB33117**	13.2 ± 1.1	1.9 ± 0.3	2.1 ± 0.3

**Table 2 ijms-25-11366-t002:** IC_50_ (μM) values of ATO and PhAs-LHP.

	ATO	PhAs-LHP
**MiaPaCa-2**	2.0 ± 0.3	0.12 ± 0.1
**PANC-1**	3.7 ± 0.6	0.21 ± 0.03
**TB33117**	1.9 ± 0.3	0.20 ± 0.05

**Table 3 ijms-25-11366-t003:** The IC_50_ (μM) values of ATO, PAO, and PhAs-LHP.

	ATO	PAO	PhAs-LHP
no amiloride	89 ± 6	1.5 ± 0.002	1.4 ± 0.08
amiloride 25 μM	95 ± 5	1.5 ± 0.05	0.7 ± 0.02

## Data Availability

The data presented in this study are available on request from the corresponding author. The data are not publicly available.

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
