# Peer review of "A Tumor Homing Peptide-Linked Arsenic Compound Inhibits Pancreatic Cancer Growth and Enhances the Inhibitory Effect of Gemcitabine"

_ijms, 2024, doi:10.3390/ijms252111366_

Round 1

Reviewer 1 Report (Previous Reviewer 1)

Comments and Suggestions for Authors

In this paper, a tumor homing peptide was linked to ATO and used in combination with gemcitabine with the aim of improving the anti-tumor effect through the targeting of this peptide. However, the article did not study the targeting effect. In addition, from the results of cell and animal experiments in this article, the antitumor effect of the peptide modification group seems not significantly improved. The authors need to compare and analyze the anti-tumor effect of the peptide modification group with unmodified group, then targeted evaluation is also needed to validate the project idea.

Author Response

In this paper, a tumor homing peptide was linked to ATO and used in combination with gemcitabine with the aim of improving the anti-tumor effect through the targeting of this peptide. However, the article did not study the targeting effect. In addition, from the results of cell and animal experiments in this article, the antitumor effect of the peptide modification group seems not significantly improved. The authors need to compare and analyze the anti-tumor effect of the peptide modification group with unmodified group, then targeted evaluation is also needed to validate the project idea.

  1. Study the targeting effect and targeted evaluation

Thank you for the critical suggestion. We have investigated the targeting effect and the mechanism involved. We are now able to show that the peptide-linked arsenic compound enters cancer cells by a macropinocytosis pathway to inhibit cancer cell proliferation as inhibition of macropinocytosis by amiloride reduced the suppression by the peptide-linked arsenic compound of cancer cell growth. We have added a new section (2.3. PhAs-LHP exerted its cytotoxic effect via macropinocytotic uptake of LHP ligand in PC cells.) to the Result on page 7 from line 174, with a new figure (Fig.5) and a new Table (Table 3) on page 7. The corresponding method was added to the Materials and Methods on page 13 from line 363 as 4.4. Investigating the cellular mechanism of Ph-As-LHP. A paragraph was added in the Discussion on page 12, lines 287-289 to address the mechanism of the peptide-linked arsenic compound, PhAs-LHP.

  1. Compare and analyze the anti-tumor effect of the peptide modification group with unmodified group and then targeted evaluation is needed to validate the project idea.

The effect of PhAs-LHP (the peptide-linked arsenic) on pancreatic cancer cell growth was compared to the non-peptide-linked arsenic compounds, ATO and PAO in vitro (Figures 3-5, Tables 2 and 3) and in vivo (xenografted mouse model of pancreatic cancer, Figures 6,7,9). The arsenic distribution and concentrations of the non-peptide-linked arsenic compound, PAO and the peptide-linked arsenic compound, PhAs-LHP were determined and presented in Figure 8 on page 10 (newly added), indicating the specific targeting of PhAs-LHP towards the pancreatic tumour. The corresponding method was added to the Materials and Methods on page 14, lines 400-405, as “4.6. GFAAS determination of arsenic distribution in mouse tissue.” and to the Supplementary Materials on page 15-16, as “Determination of arsenic distribution and concentration by GFAAS”.

Reviewer 2 Report (New Reviewer)

Comments and Suggestions for Authors

Interesting study. I only consider that it would be relevant to evaluate in animals the toxicity of the the peptide-linked arsenic compound in other evaluations other than "any significant decrease in mouse body weight".

Additional comments:

The potential application of a peptide-linked arsenic compound already prepared and described (J Med Chem, 66 (2023) 12101-12114) in pancreatic cancer treatment, evaluated by in vitro and in vivo studies. This is a relevant topic, because the treatments for  pancreatic cancer are very limited and mostly palliative. The present work explored, for the first time (as far as I could see in bibliography analysis) the potential application of a peptide-linked arsenic compound in this type of cancer and the results are stimulating. Methodology seems ok. I only consider that it would be relevant to evaluate in animals the toxicity of the the peptide-linked arsenic compound in other evaluations other than "any significant decrease in mouse body weight". The conclusions consistent with the evidence and arguments and the references are appropriate.

Comments on the Quality of English Language

Minor editing of English language required.

Author Response

Interesting study. I only consider that it would be relevant to evaluate in animals the toxicity of the the peptide-linked arsenic compound in other evaluations other than "any significant decrease in mouse body weight".

Additional comments:

The potential application of a peptide-linked arsenic compound already prepared and described (J Med Chem, 66 (2023) 12101-12114) in pancreatic cancer treatment, evaluated by in vitro and in vivo studies. This is a relevant topic, because the treatments for  pancreatic cancer are very limited and mostly palliative. The present work explored, for the first time (as far as I could see in bibliography analysis) the potential application of a peptide-linked arsenic compound in this type of cancer and the results are stimulating. Methodology seems ok. I only consider that it would be relevant to evaluate in animals the toxicity of the the peptide-linked arsenic compound in other evaluations other than "any significant decrease in mouse body weight". The conclusions consistent with the evidence and arguments and the references are appropriate.

Evaluate in animals the toxicity of the peptide-linked arsenic compound in other evaluations other than "any significant decrease in mouse body weight".

Thank you for the positive and constructive comments. We have taken the suggestion to investigate further the in vivo toxicity of the non-peptide-linked arsenic compound, PAO and the peptide-linked arsenic compound, PhAs-LHP by determination of arsenic distribution and concentrations of PAO and PhAs-LHP. We have obtained new data to show that PAO had twice high concentration in the liver than in the heart without any detectable level in the brain and kidney, and that PhAs-LHC did not have any detectable level of arsenic in any of the above organs mentioned, suggesting the specific targeting of PhAs-LHP towards the pancreatic tumour. This data has been added as Figure 8 on page 10. The corresponding method was added to the Materials and Methods on page 14, lines 400-405, as “4.6. GFAAS determination of arsenic distribution in mouse tissue.” and to the Supplementary Materials on page 15-16, as “Determination of arsenic distribution and concentration by GFAAS”.

Round 2

Reviewer 1 Report (Previous Reviewer 1)

Comments and Suggestions for Authors

The author made some changes to the manuscript. However, from the results of the latest version of the experimental data, we did not see a significant improvement in the anti-tumor effect  after a tumour homing peptide modification. From the results, the use of peptide modification is of little significance.

Author Response

Reviewer No.1

The author made some changes to the manuscript. However, from the results of the latest version of the experimental data, we did not see a significant improvement in the anti-tumor effect  after a tumour homing peptide modification. From the results, the use of peptide modification is of little significance.

Thank you for this critical comment. The peptide-linked arsenic compound (PhAs-LPH), at 2-times lower concentration than its non-peptide-linked analogue (PAO), showed similar efficacy of inhibiting pancreatic tumor growth as PAO did. The 2-times difference may be minimal as the reviewer pointed out. The reduced toxicity from PhAs-LHP demonstrated by significantly less arsenic distribution in the heart and liver, indicates the potential for further improving the anti-tumor efficacy by increasing the dosage of PhAs-LHP. Further modification of the peptide-linked arsenic compound with increased efficacy and reduced toxicity will contribute to the clinical application of the arsenic-based chemotherapy. To address this point, we have added:

  1. A paragraph “The reduction in tumor volume was similar between PAO+Gem and PhAs-LHP+Gem although the tumor weight of the mice treated with PhAs-LHP showed minimal but significant reduction compared to PAO+Gem (Fig. 7). The arsenic distribution in the heart was much lower in the PhAs-LHP+Gem treated mice than in the PAO+Gem treated mice (Fig. 8), leading to a potential reduction in cardiotoxicity which is an adverse effect observed with arsenic chemotherapy.” in page 12, the 4th
  2. These sentences “PhAs-LHP, at 2 times less molar concentration than PAO, reduced tumor growth to a similar level by PAO. The 2-times difference may be minimal. However, the fact that a lot less arsenic distribution in the heart and liver from the PhAs-LHP-treated mice than from the PAO-treated mice, indicated a reduced toxicity from PhAs-LHP.” to the 5th paragraph in page 12.
  3. To address the limitation of the study and future investigation, a paragraph of “PhAs-LPH, at 2-times lower concentration than PAO, showed similar efficacy of inhibiting pancreatic tumor growth as PAO did. The 2-times difference may be minimal. The reduced toxicity from PhAs-LHP demonstrated by significantly less arsenic distribution in the heart and liver, indicates the potential for further improving the anti-tumor efficacy by increasing the dosage of PhAs-LHP. Modification of the peptide-linked arsenic compound with increased efficacy and reduced toxicity will promote the clinical application of the arsenic-based chemotherapy.” was added in page 12 as the last paragraph.

This manuscript is a resubmission of an earlier submission. The following is a list of the peer review reports and author responses from that submission.

Round 1

Reviewer 1 Report

Comments and Suggestions for Authors

In this paper, a targeted peptide was linked to ATO and used in combination with gemcitabine with the aim of improving the anti-tumor effect. However, this paper did not carry out relevant research on the synergistic mechanism of drug combination,such as analysis of ATO down-regulated Skp2 proteins. In addition, PhAs-LHP synthesis methods, structural characterization experiments and PhAs-LHP targeted uptake experiment are lacking. It is suggested that Figures 5 and 6 be analyzed and discussed together.

Author Response

Thank you for the constructive comment which help to improve our paper significantly. We have addressed your concerns as below.

In this paper, a targeted peptide was linked to ATO and used in combination with gemcitabine with the aim of improving the anti-tumor effect. However, this paper did not carry out relevant research on the synergistic mechanism of drug combination such as analysis of ATO down-regulated Skp2 proteins. In addition, PhAs-LHP synthesis methods, structural characterization experiments and PhAs-LHP targeted uptake experiment are lacking. It is suggested that Figures 5 and 6 be analyzed and discussed together.

  1. relevant research on the synergistic mechanism of drug combination such as analysis of ATO down-regulated Skp2 proteins

We agree with the reviewer’s comment on the relevant research on the synergistic mechanism involved in the combination effect. It has been reported that ATO inhibited pancreatic cancer (PC) and enhanced the inhibitory effects of gemcitabine and the hypoxia-inducible factor 1 inhibitor PX-478 by reducing proliferation and increasing apoptosis. PhAs-LHP, the peptide-linked arsenic compound, is expected to have significantly increased the effect of ATO by increasing its entry into cancer cells via macropinocytotic pathways (Carrall JA, J Med Chem 2023, 66(17):12101-12114). We have shown in this paper that ATO and PhAs-LHP decreased the expression of Skp2 in pancreatic tumours with or without gemcitabine, indicating that ATO and PhAs-LHP inhibit PC growth by affecting the Skp2 signaling pathway. In the future, we will carry out a further study to investigate how ATO and PhAs-LHP suppressed the Skp2 protein and its downstream pathway.

  1. PhAs-LHP characterization

The synthesis, structural characterization, and targeted uptake of PhAs-LHP are fully described and published in one of our previous papers (Carrall JA, J Med Chem 2023, 66(17):12101-12114). We apologize that we did not describe this clearly in the manuscript and we have now added a line in the Introduction, line 76 and in the Materials and Methods, line 267.

  1. Combine the presentation of Fig.5 and Fig.6

We have taken the suggestion of the review and analyzed the data in Figures 5 and 6 together by combining results 2.3 to 2.4 under a new title: 2.3. Arsenic-peptide suppressed pancreatic cancer growth and enhanced the inhibitory effect of gemcitabine on pancreatic cancer growth. We have changed the title of results section 2.4 to Arsenic-peptide suppressed pancreatic cancer growth by downregulation of Skp2. 

Reviewer 2 Report

Comments and Suggestions for Authors

In this manuscript “A Tumor Homing Peptide-Linked Arsenic Compound Inhibits Pancreatic Cancer Growth and Enhances the Inhibitory Effect of Gemcitabine”, there are a number of issues as follows:

1.        Its hard to believe how the authors are coming to a biological conclusion with an in vivo experiment with having just 4 animals as shown in Fig 6.

2.       There is no mechanistic evidence of the finding, about how the Ato-LHP is improving the therapeutic regimen.

3.       Why in Fig 6, ~4 animals were used while in Fig5, ~ 8 animals were used?

4.       The systemic effect of this Ato-LHP on murine Hematopoietic, Bilirubin, Creatinine should have been thoroughly discussed to understand how better is the combination with respect to Ato alone.

5.        When the authors give this type of title, then the effect of PhA-LHP, Ato alone effect on gemcitabine regimen should have been discussed.

6.        Authors have not described how they prepared this compound. How stable it is in the in vivo setting, nothing is mentioned. These appear to be dubious and raises question over authenticity.

7.       Authors should have done orthotopic implantation instead of subcutaneous which often donot represent the correct PDAC biology.

8.       Why the authors didn’t use the B6 model they tried earlier in PMID: 33126623?

9.       ithenticate shows this manuscript has 31% match, this has to be reduced significantly.

10.   Why Skp2 staining was done in not justified properly. It seems a cherry-picked approach. Authors should do cleaved caspase, Ki67 staining.

Author Response

Thank you for the critical and constructive comments which we have addressed point by point as below.

In this manuscript “A Tumor Homing Peptide-Linked Arsenic Compound Inhibits Pancreatic Cancer Growth and Enhances the Inhibitory Effect of Gemcitabine”, there are a number of issues as follows:

  1. Its hard to believe how the authors are coming to a biological conclusion with an in vivo experiment with having just 4 animals as shown in Fig 6.

From our previous publication (Can Lett 346 (2014) 264-272), 4-5 mice in each group for testing the effect of gemcitabine is sufficient to obtain statistically significant data. According to our experience of testing small molecular compounds in such a xenografted mouse model, we initially used 8 mice per group to test the effects of arsenic (PAO) and its peptide-linked compound (PhAs-LHP) on tumour growth. Once it was confirmed that the variations in the effects of either PAO or PhAs-LHP on the growth of xenografted tumours were low, as demonstrated in Fig.5, we had used reduced numbers of mice in the experiments testing the combination effects as shown in Fig.6.

  1. There is no mechanistic evidence of the finding, about how the Ato-LHP is improving the therapeutic regimen.

The detailed mechanism of how PhAs-LHP improves the therapeutic effect has been demonstrated and discussed in our previous publication (Carrall JA, J Med Chem 2023, 66(17):12101-12114), where it has been shown that PhAs-LHP increased the cellular uptake of the arsenic compound most likely via micropinocytosis pathway, which is upregulated in many cancer cells, including pancreatic cancer cells. This point has now been now added to the second last paragraph in the Discussion section, lines 247-250, on page 10.

  1. Why in Fig 6, ~4 animals were used while in Fig5, ~ 8 animals were used?

Refer to the answer to question 1.

  1. The systemic effect of this Ato-LHP on murine Hematopoietic, Bilirubin, Creatinine should have been thoroughly discussed to understand how better is the combination with respect to Ato alone.

We agree with the reviewer on this constructive suggestion to compare the systemic effect of ATO and its peptide-linked compound, which will be particularly important for the clinical translation of the findings. In fact, our findings in this paper warrant further study to compare the systemic effect of ATO and its peptide-linked compound, especially their toxicities. These important points have been added to the last paragraph of the Discussion, on page 10, lines 257-259.

  1. When the authors give this type of title, then the effect of PhA-LHP, Ato alone effect on gemcitabine regimen should have been discussed.

The data of comparing the effects of ATO and PhAs-LHP on pancreatic tumour growth with or without gemcitabine was shown in the supplementary Fig.2 under the Supplementary Materials. The description of the results was in the last paragraph of the Result section 2.3, the last sentence, lines 186-187 on page 8.

  1. Authors have not described how they prepared this compound. How stable it is in the in vivo setting, nothing is mentioned. These appear to be dubious and raises question over authenticity.

The synthesis, structural characterization, stability studies, targeted uptake, and cytotoxicity of PhAs-LHP have been investigated and published in one of our previous papers (Carrall JA, J Med Chem 2023, 66(17):12101-12114). We apologize that we did not describe this clearly in the manuscript and we have now added a line in the Introduction, line 76 and in the Materials and Methods, line 267.

  1. Authors should have done orthotopic implantation instead of subcutaneous which often donot represent the correct PDAC biology.

Thank you for this constructive suggestion. As this is only an initial study where we aimed to test the hypothesis that a peptide-linked arsenic compound will supress pancreatic cancer growth and enhance the inhibitory effect of gemcitabine at a lower concentration with reduced toxicity and increased efficacy, we have selected the subcutaneous flank tumour model which allows us to obtain enough data to test the hypothesis. The orthotopic models represent a condition closer to PDA, and will be used for further investigation to draw a complete conclusion.

  1. Why the authors didn’t use the B6 model they tried earlier in PMID: 33126623?

       The C57BL6 syngeneic mouse model is an immune competent model used to investigate the immune-related effects. Since the aim of this study was to determine if the peptide-linked arsenic compound, PhAs-LHP had an increased anti-cancer effect compared to ATO by increasing the intra-cellular amount of effective ATO via macropinocytotic-type pathways (Carrall JA, J Med Chem 2023, 66(17):12101-12114), independently of any effects on the immune system, this Scid xenografted model served this purpose well. In future studies it will also be interesting to investigate the effects of ATO and PhAs-LHP on tumour immune response in the C57BL6 mouse.

  1. ithenticate shows this manuscript has 31% match, this has to be reduced significantly.

       We have carefully edited and made many changes to the manuscript. Hopefully this would reduce the similarity area significantly.

  1. Why Skp2 staining was done in not justified properly. It seems a cherry-picked approach. Authors should do cleaved caspase, Ki67 staining.

We apologize for this unclear justification for Skp2 staining. We have added a statement in the Results section 2.4 to explain more clearly the rationale to determine the Skp2 expression in those tumours with extra references added on page 9, lines 195-199.

As previous similar studies have demonstrated that ATO inhibited pancreatic cancer growth and enhanced the inhibition by gemcitabine via decreasing proliferation (by Ki67 staining) and increased apoptosis (by the staining of active caspase 3) (Cancer Manag Res 2020, 12:13149-13159; Transl Res 2023, 255:66-76) in a similar xenografted mouse model, we have chosen to cite those results without repeating the assays of Ki67 and caspase 3. However, the expression Skp2 was only reported in cells, not in tumour tissue. To explore the mechanisms involved in the effects ATO and its peptide-linked compound on tumour growth, the expression of Skp2 in tumour tissues was determined.

Reviewer 3 Report

Comments and Suggestions for Authors

In the study “A Tumor Homing Peptide-Linked Arsenic Compound Inhibits Pancreatic Cancer Growth and Enhances the Inhibitory Effect of Gemcitabine” the authors aimed to determine whether coupling arsenic to a tumor homing peptide would increase the inhibitory potency against pancreatic cancer cells. This is extremely important as Arsenic trioxide is highly toxic. It has been shown that the peptide-linked arsenic compound inhibited pancreatic cancer more potently than Arsenic trioxide. I find this study very interesting and addresses the audience in the field of oncology, chemical biology and therapeutics. 

The manuscript topic is relevant. However, I have some concerns and need further clarifications to consider this manuscript to be published in IJMS.

  1. Although authors mentioned use of PF-3758309 in Figure 2, did the authors try any other drug in addition to gemcitabine? For example Capecitabine and nab-paclitaxel. Readers can be curious as only gemcitabine is used in combination with PhAs-LHP and other ATO analogs. 

  2. As the authors highlighted PAK1 and PAK4 kinases as well as KRAS are mutated in a significant number of pancreatic cancer cases, did the authors observe direct effect on these proteins upon treatment with PAO, PhAs-LHP and gemcitabine?

  3. Page 6, lines 163-164: the authors state that “The doses of PAO and PhAs-LHP were calculated from the results obtained from the in vitro assay”. I am not sure how accurate this approximation is. In-vitro and in-vivo/in-cell conditions can be very different due to the heterogeneous and crowded environment of cells. Authors may provide further insights to address this question.

  4. Lines 43-53, more references can be added. Similarly lines 96-108, certain references to PAK kinases are missing.

  5. Figure 5 caption is too short as individual panels are not well described. In addition what CT stands for is also not clear, although mentioned somewhere in the text.

  6. Certain figure panels are confusing, for example Figs. 5A, 5D, 6A, 6D, due to error bars on top of the points. It would be better for the readers if colored lines are used for each condition.

  7. I recommend the authors for a careful reading of the main text and insert appropriate references wherever missing. I found several paragraphs where certain sentences lack citations. Please update the list of references.

Author Response

In the study “A Tumor Homing Peptide-Linked Arsenic Compound Inhibits Pancreatic Cancer Growth and Enhances the Inhibitory Effect of Gemcitabine” the authors aimed to determine whether coupling arsenic to a tumor homing peptide would increase the inhibitory potency against pancreatic cancer cells. This is extremely important as Arsenic trioxide is highly toxic. It has been shown that the peptide-linked arsenic compound inhibited pancreatic cancer more potently than Arsenic trioxide. I find this study very interesting and addresses the audience in the field of oncology, chemical biology and therapeutics. 

The manuscript topic is relevant. However, I have some concerns and need further clarifications to consider this manuscript to be published in IJMS.

Thank you for your positive comments on the impact of our paper on the field of oncology, chemical biology and therapeutics. Thank you for your constructive suggestions which surely help us to improve the manuscript significantly. We have addressed all questions raised point by point as below. We hope that we have answered the questions properly to make the manuscript satisfy the requirements for publication.

  1. Although authors mentioned use of PF-3758309 in Figure 2, did the authors try any other drug in addition to gemcitabine? For example Capecitabine and nab-paclitaxel. Readers can be curious as only gemcitabine is used in combination with PhAs-LHP and other ATO analogs. 

It is interesting to investigate the combination of this peptide-linked arsenic compound, PhAs-LHP with other drugs such as capecitabine and nab-paclitaxel mentioned by the reviewer. This will be included in further study.

2. As the authors highlighted PAK1 and PAK4 kinases as well as KRAS are mutated in a significant number of pancreatic cancer cases, did the authors observe direct effect on these proteins upon treatment with PAO, PhAs-LHP and gemcitabine?

The effect of gemcitabine on PAK1 and PAK4 expression was reported by us previously (BMC Cancer. 2016 Jan 16;16:24; Am J Physiol Gastrointest Liver Physiol. 2019 May 1;316(5):G632-G640). It is highly interested in testing the effects of PAO and PhAs-LHP on PAK1, PAK4 and KRas in pancreatic cancer, which shall be included in future study when testing the combination of PAO and PhAs-LHP with multiple drugs in pancreatic cancer.

3. Page 6, lines 163-164: the authors state that “The doses of PAO and PhAs-LHP were calculated from the results obtained from the in vitro assay”. I am not sure how accurate this approximation is. In-vitro and in-vivo/in-cell conditions can be very different due to the heterogeneous and crowded environment of cells. Authors may provide further insights to address this question.

Thank you for this critical point. Referring to the IC 50 values obtained from the in vitro assays, we had calculated a range of dosage for both PAO and PhAs-LHP. The final dosages for PAO and PhAs-LHP were selected after a toxicity test in mice. We have changed the statement in line 149, on page 6 to describe it more accurately.

4. Lines 43-53, more references can be added. Similarly lines 96-108, certain references to PAK kinases are missing.

We apologize for these careless mistakes. We have added references as required. References No. 8 and 9 were added in line 46; reference No.21 in line 95, reference No.22 in line 96.

5. Figure 5 caption is too short as individual panels are not well described. In addition what CT stands for is also not clear, although mentioned somewhere in the text.

We have added more detail to the legend of Fig.5 and described each penal properly as the reviewer suggested.

6. Certain figure panels are confusing, for example Figs. 5A, 5D, 6A, 6D, due to error bars on top of the points. It would be better for the readers if colored lines are used for each condition.

According to the reviewer’s suggestion, we have changed lines in Figures 5A, 5D, 6A and 6D to different colors.

7. I recommend the authors for a careful reading of the main text and insert appropriate references wherever missing. I found several paragraphs where certain sentences lack citations. Please update the list of references.

Thank you for the detailed comments and suggestions. We apologize for these careless mistakes. We have carefully read the manuscript and added references wherever it is missing. Refer to the answer to question 4. In addition, Reference No.19 was added in line 71, on page 2. Reference No. 9 was added in line 244, on page 10.

Round 2

Reviewer 1 Report

Comments and Suggestions for Authors

This paper seems to be a small part of the project, the main content of the research has been published(Carrall JA, J Med Chem 2023, 66(17):12101-12114). This paper provides only limited research data.

Reviewer 2 Report

Comments and Suggestions for Authors

Suggested in vivo based confirmatory assays are missing.

Reviewer 3 Report

Comments and Suggestions for Authors

I appreciate the effort of the authors to incorporate my suggestions. By including the suggested modifications, the authors have addressed my major concerns and suggestions. This is now a reasonably good article and I support the publication of this manuscript in IJMS.